# Early Enriched Environment Prevents Epigenetic p11 Gene Changes Induced by Adulthood Stress in Mice

**DOI:** 10.3390/ijms22041928

**Published:** 2021-02-15

**Authors:** Mi Kyoung Seo, Ah Jeong Choi, Dae-Hyun Seog, Jung Goo Lee, Sung Woo Park

**Affiliations:** 1Paik Institute for Clinical Research, Inje University, Busan 47392, Korea; banaba66@inje.ac.kr; 2Sumagen, Seoul 06159, Korea; dkwjd2675@naver.com; 3Department of Biochemistry, College of Medicine, Inje University, Busan 47392, Korea; daehyun@inje.ac.kr; 4Dementia and Neurodegenerative Disease Research Center, Inje University, Busan 47392, Korea; 5Department of Psychiatry, College of Medicine, Haeundae Paik Hospital, Inje University, Busan 48108, Korea; 6Department of Health Science and Technology, Graduate School, Inje University, Busan 47392, Korea; 7Department of Convergence Biomedical Science, College of Medicine, Inje University, Busan 47392, Korea

**Keywords:** chronic unpredictable stress, depression, early enriched environment, epigenetic mechanism, p11

## Abstract

Positive experiences in early life may improve the capacity to cope with adulthood stress through epigenetic modification. We investigated whether an enriched environment (EE) in the postnatal period affected epigenetic changes in the p11 gene induced by chronic unpredictable stress (CUS) in adult C57BL/6J mice. EE was introduced for 5 weeks during postnatal days 21–55. After EE, the mice were subjected to CUS for 4 weeks. EE prevented depression-like behavior induced by adult CUS. EE prevented a decrease in p11 mRNA and histone H3 acetylation induced by CUS, with changes in the expression of histone deacetylase 5. Moreover, EE prevented changes in trimethylation of histone H3 lysine 4 (H3K4) and H3K27 induced by CUS. Furthermore, EE had positive effects on behavior and epigenetic alterations in adult mice without CUS. These results suggest that one of the underlying mechanisms of early-life EE may involve epigenetic modification of the hippocampal p11 gene promoter.

## 1. Introduction

Human responses to stress vary among individuals. Some people develop stress-related psychiatric disorders such as depression, others develop mild to moderate psychiatric symptoms that recover quickly, and still others report no symptoms in response to stress [1,2]. Interindividual difference in vulnerability and resilience to stress may depend on early life experiences. Positive experiences in early life may improve the capacity to cope with adulthood stress through epigenetic modification. Numerous studies have indicated that adverse experiences in early life augment the risk of depression in adulthood and that this effect is influenced by epigenetic mechanisms [3,4,5]. Compared to numerous studies on long-term effects of negative experiences in early life, there is limited research on the effect of positive experiences in early life. In rodents, high levels of maternal care exhibited long-term effects on the epigenomic state, brain function, and behavior of offspring [6,7].

Posttranslational modification of histones (i.e., histone acetylation and methylation), a well-studied epigenetic mechanism, is a process that controls gene activation or silencing by altering the chromatin structure and compaction in the promoter regions of particular genes [8,9]. Histone acetylation (H3 and H4 acetylation) relaxes the interaction between DNA and histone, allowing the transcriptional machinery to bind to the promoters of particular genes, thereby activating transcription [8,9]. Coactivator complexes needed for transcriptional activation have histone acetyltransferases (HATs), whereas transcriptional corepressor complexes have histone deacetylases (HDACs) [10,11]. Histone methylation, on the other hand, can influence both transcriptional activation (H3K4 and H3K36) and repression (H3K9, H3K27, and H3K20), depending on the lysine (K) residue of the histone tail as well as the valence state of the methylation (mono-, di-, or trimethylation) [12,13].

Expression of p11, also called S100A10, plays a critical role in depression-like behaviors and responses to antidepressant drugs [14]. For example, p11 expression was decreased in the brains of humans with depression and in mouse models of depression, and it was increased by treatment with antidepressant drugs [15]. Moreover, p11 knockout mice exhibited depression-like behaviors, and transgenic mice overexpressing p11 exhibited antidepressant-like effects [15,16]. Furthermore, p11 expression is regulated by brain-derived neurotropic factor (BDNF), which plays important roles in neural plasticity and neurogenesis in the hippocampus, and the antidepressant-like effects of BDNF depend on p11 [17]. 

An enriched environment (EE) is the manipulation of standard laboratory conditions to reach optimization of the breeding environment by modifying the quality and intensity of environmental stimuli. EE is also known to have a significant effect on the central nervous system (CNS) at a functional, anatomical, and molecular level both in the critical period of development and in adulthood [18]. It is known that EE has an effect on neural plasticity of the brain and the possibility of it becoming a new target for the treatment of neurological and mental disorders has been suggested [19]. In general, in animal experiments, EE comprises housing conditions that provide sensory, cognitive, and motor stimulation [20]. EE treatment enhanced neurogenesis and synaptic plasticity and increased BDNF expression in the hippocampus [21]. In rats, EE exposure after weaning reduced adult anxiety-like behavior [22] and influenced hippocampal gene expression [23,24]. In particular, EE in rats was effective in protecting from the negative effects of stressors [22,25]. These studies highlight the antidepressant effects of EE. In this study, we focused on the long-term effects of early-life EE on resilience under adulthood stress. Therefore, we investigated whether early EE after weaning can induce long-lasting effects on depression-like behavior induced by adulthood stress and whether the effects of EE are involved in epigenetic modification of the p11 gene. According to “neurotrophic hypothesis of depression”, stress-induced changes in hippocampal gene expression are known to play an important role in the pathophysiology of depression [26,27]. EE improves the neural plasticity of the hippocampus. Thus, we examined the molecular levels in the hippocampus. We employed chronic unpredictable stress (CUS), an animal model of depression, as a form of adult stress.

## 2. Results

### 2.1. Effects of EE and CUS on Depression-Like Behavior and Plasma Corticosterone Levels

The experimental design is shown in Figure 1. First, we examined the effects of CUS on depression-like behavior and corticosterone levels in mice housed under standard or EE conditions during early life (Figure 2). 

Immobility times during the forced swimming test (FST) (*n* = 4–6 animals per group) are presented in Figure 2A. Two-way ANOVA revealed the following values: EE *F*_(__1,15)_ = 7.158, *p* = 0.017; CUS *F*_(1,15)_ = 33.889, *p* < 0.001; EE × CUS interaction *F*_(4,53)_ = 33.889, *p* < 0.001. Post hoc comparisons showed that CUS-exposed mice exhibited a significant depression-like phenotype compared to controls (90.3 ± 9.7 vs. 137.3 ± 14.4 s, *p* = 0.040). However, exposure to EE significantly decreased the immobility time in CUS-treated mice (137.3 ± 14.4 vs. 33.1 ± 12.0 s, *p* < 0.001), suggesting that early-life EE prevented depression-like behavior induced by CUS in adult mice. Moreover, EE mice displayed reduced immobility time compared to controls (90.3 ± 9.7 vs. 40.2 ± 4.4 s, *p* = 0.010), showing that treatment with EE alone during early life had an antidepressant effect in adulthood. 

Figure 2B shows the effects of EE and CUS on plasma corticosterone levels (*n* = 12–15 animals per group). Two-way ANOVA revealed significant effects of EE: *F*_(__1,50)_ = 8.463, *p* = 0.005; and CUS, *F*_(1,50)_ = 10.960, *p* = 0.002; but no EE × CUS interaction effect, *F*_(1,50)_ = 2.177, *p* = 0.146. CUS exposure led to a significant increase in corticosterone levels compared to control conditions (9120 ± 692 vs. 12,317 ± 836 pg/mL, *p* = 0.010), and exposure to EE prevented the CUS-induced increase in corticosterone level (12,317 ± 836 vs. 9388 ± 733 pg/mL, *p* = 0.020). 

### 2.2. Effects of EE and CUS on Hippocampal p11 mRNA Levels

The effect of EE and CUS on p11 mRNA levels in the hippocampus was determined by qRT-PCR, as shown in Figure 3 (*n* = 12–15 animals per group). Two-way ANOVA revealed significant effects of EE: *F*_(__1,50)_ = 30.300, *p* < 0.001; CUS *F*_(1,50)_ = 88.110, *p* < 0.001; and the EE × CUS interaction *F*_(1,50)_ = 7.464, *p* = 0.009. Post hoc analysis revealed a significant decrease in p11 mRNA levels in CUS-exposed mice compared to control mice (100 ± 8% vs. 39 ± 3%, *p* < 0.001), whereas early-life EE prevented this reduction (39 ± 3% vs. 74 ± 3%, *p* = 0.003). Additionally, the p11 mRNA level was 39% higher in EE-exposed mice compared with control animals (100 ± 8% vs. 139 ± 9%, *p* < 0.001). 

### 2.3. Effects of EE and CUS on the Epigenetic State of the p11 Gene

We next evaluated whether postnatal exposure to EE induced histone modification that led to activation or repression of the p11 gene. We examined two active histone modifications (acetylation of histone H3, AcH3; trimethylation of H3K4, H3K4me3) and one repressive histone modification (trimethylation of H3K27, H3K27me3) in the p11 promoter region in the hippocampus (*n* = 12–15 animals per group). 

Two-way ANOVA on AcH3 levels (Figure 4A) revealed significant effects of EE: *F*_(__1,50)_ = 57.530, *p* < 0.001; CUS *F*_(1,50)_ = 16.980, *p* < 0.001; and the EE × CUS interaction *F*_(1,50)_ = 3.099, *p* = 0.084. CUS reduced the AcH3 level at the p11 promoter compared to the control condition (100 ± 11% vs. 60 ± 3%, *p* = 0.037), whereas EE increased this level in CUS mice (60 ± 3% vs. 134 ± 6%, *p* = 0.018). EE alone significantly increased AcH3 compared to control mice (100 ± 11% vs. 174 ± 14%, *p* < 0.001). Then, we further examined whether the candidate enzyme HDAC5 was associated with histone acetylation changes observed in the hippocampus. Two-way ANOVA on HDAC5 mRNA levels (Figure 4B) revealed the following values: EE *F*_(__1,50)_ = 41.830, *p* < 0.001; CUS *F*_(1,50)_ = 254.900, *p* < 0.001; and the EE × CUS interaction *F*_(1,50)_ = 4.585, *p* = 0.037. Post hoc analysis showed that HDAC5 mRNA levels were 335% higher in the hippocampus of CUS-exposed mice compared with control animals (100 ± 5% vs. 335 ± 17%, *p* < 0.001), whereas EE exposure prevented this effect of CUS on HDAC5 expression (335 ± 17% vs. 223 ± 19%, *p* < 0.001). Moreover, EE decreased hippocampal HDAC5 expression compared to control conditions (100 ± 5% vs. 44 ± 3%, *p* = 0.015). 

Two-way ANOVA on H3K4me3 levels (Figure 5A) revealed the following values: EE *F*_(__1,50)_ = 27.860, *p* < 0.001; CUS *F*_(1,50)_ = 21.130, *p* < 0.001; and the EE × CUS interaction *F*_(1,50)_ = 0.161, *p* = 0.690. Exposure to CUS induced a significant decrease in the H3K4me3 level compared to control mice (100 ± 9% vs. 50 ± 6%, *p* = 0.007), and EE restored the H3K4me3 level at the p11 promoter in the hippocampus of CUS-exposed animals to normal levels (50 ± 6% vs. 107 ± 11%, *p* = 0.017). A significant difference in the H3K4me3 level was observed between control and EE-exposed animals (100 ± 9% vs. 149 ± 12%, *p* = 0.005). 

Two-way ANOVA on H3K27me3 levels (Figure 5B) revealed the following values: EE *F*_(__1,50)_ = 50.590, *p* < 0.001; CUS *F*_(1,50)_ = 109.700, *p* < 0.001; and the EE × CUS interaction *F*_(1,50)_ = 5.181, *p* = 0.027. The level of H3K27me3 was increased at the p11 promoter in CUS-exposed mice compared with control mice (100 ± 7% vs. 222 ± 17%, *p* < 0.001). Importantly, early-life EE significantly decreased the H3K27me3 level at the p11 promoter in CUS-exposed mice (222 ± 17% vs. 132 ± 8%, *p* < 0.001). Moreover, a significant effect of EE was observed in comparison with the control mice (100 ± 7% vs. 54 ± 4%, *p* = 0.006).

## 3. Discussion

Early enrichment experience may promote offspring brain health and increase early mental development [28]. EE is already well known for its positive effects on depression-like and anxiety-like behaviors in human and animal models. However, the mechanism of early EE’s beneficial effect on adulthood depression has not been investigated. In the present study, EE prevented the depression-like behavior and increased corticosterone observed in a CUS model of depression. One of the underlying mechanisms of early-life EE may involve epigenetic modification of the hippocampal p11 gene promoter.

Our behavioral data demonstrate that early EE prevented the behavioral despair state induced by CUS during adulthood, suggesting that early EE improves the ability to cope with stress under an unpredictable stress situation, as in CUS. Moreover, exposure to EE during PND 21–55 exerted antidepressant-like effects similar to those of antidepressant drugs when animals were tested at PND 84. Our data demonstrate for the first time that the long-term effects of early EE resulted from increased p11 expression accompanied by alterations in histone modification within the p11 promoter. 

In this experiment, CUS animals exhibited higher levels of stress-induced corticosterone secretion compared to control animals. This result is consistent with previous reports showing reduced body weight gain, hypercorticosteronemia, increased adrenal weight, and anxiety- and depression-like behaviors [29,30]. In previous study, mice exposed to chronic restraint stress exhibited reduced p11 expression in medial prefrontal cortex and depression-like behaviors [31]. In the results of the functional analysis of p11, p11 overexpression in neurons of the medial prefrontal cortex rescued depression-like behaviors induced by chronic stress, and further alleviated depression-like phenotypes exhibited in neuron-specific or global deleted mice of p11 [31]. They emphasized that p11 in neurons is a key molecule in chronic stress-induced depression. Consistent with previous studies, our study also showed that reduced p11 expression influenced CUS-induced behavior. To strengthen this point, further research is needed on the functional analysis of p11. We performed the FST to assess the depression-like behavior. Although many researchers use this model for behavioral despair tests, the FST is a more appropriate tool for screening antidepressant drugs [32]. Additional behavioral tests are needed, such as the sucrose preference test to measure anhedonia and the Morris water maze test to measure spatial memory.

In our experiment, CUS-induced depressive mice displayed significantly decreased acetylation of H3 (K9 and K14) at the p11 promoter of the hippocampus with obviously increasing HDAC5 expression. In agreement with these results, Liu et al. reported that rats exposed to CUS exhibited significant decreases in acetylation of H3 (K9) and H4 (K12) with increasing HDAC5 in the hippocampus, which was linked to anxiety- and depression-like behavior [33]. Erburu et al. found that chronic social defeat stress induced a decrease in the acetylation levels of H3 and H4 at the BDNF promoter as well as an increase in nuclear HDAC5 expression [34]. Similarly, chronic variable stress resulted in decreased histone acetylation of H3 and H4 and increased expression of the HDAC5 protein, which was related to lower weight, a higher corticosterone level, higher adrenal weight, and lower sucrose preference [35]. These studies suggest that the hypoacetylation of histone induced by chronic stress may cause changes in chromatin structure and compaction and modulate transcription of a depression-related gene such as p11, as observed in the present study. 

It is generally believed that an increase in HDACs would cause a decrease in the levels of histone acetylation, leading to gene silencing. Reduced HDAC expression would increase gene transcription [36]. Expression of HDAC5, but not of several other HDACs, was downregulated in the hippocampus of animals exposed to chronic stress that had been treated with antidepressant drugs [37,38]. Consistent with the present study, CUS increased HDAC5 expression in the rat hippocampus [33]. HDAC5 mRNA levels were increased in the peripheral leukocytes of drug-free depressive patients, and this increase was recovered by paroxetine treatment [39]. Moreover, HDAC5 was reported to epigenetically control behavioral adaptation to chronic emotional stimuli [40]. Based on these studies, we focused specifically on HDAC5 among the classes of HDACs. However, a weakness of the study was that the expression of other HDAC isoforms (HDAC1-11) was not examined. Thus, further studies on other HDAC isoforms associated with depression are needed to demonstrate the specificity of changes in HDAC5 expression. 

Additionally, we observed significant changes in histone methylation of K4, a permissive methylation marker, and K27, a repressive methylation marker. Chronic stress regulation of histone methylation in the hippocampus has been investigated in several studies. Chronic social defeat stress produced a robust increase in H3K27 dimethylation at the BDNF promoter, whereas H3K9 dimethylation, another histone modification that is correlated with transcriptional repression, was not increased [37]. In that study, social defeat stress did not affect the dimethylation of H3K4, but chronic imipramine treatment was able to increase this modification in mice exposed to defeat stress [37]. In another study, chronic restraint stress increased global levels of H3K9me3 and H3K4me3 in the hippocampus, whereas global levels of H3K27me3 showed no effect [41]. Our study showed that CUS decreased H3K4me3 and increased H3K27me3 at the p11 promoter of the hippocampus. It appears that modes of histone methylation may differ based on methylation of specific gene promoters vs. global methylation. Further studies are necessary to understand the functional importance of these various modes of histone methylation in the hippocampus.

In the present study, early EE had a strong antidepressant effect in adulthood. Preclinical studies using animal models of depression have shown that EE reverses depression-like behavior and improves cognitive impairment by exposure to stress (i.e., early life stress, prenatal chronic stress, adult stress) [25,42,43]. EE markedly reversed a prolonged corticosterone secretion induced by prenatal stress [44]. EE has been shown to exert antidepressant effects by restoring reduced BDNF, nerve growth factor, neurogenesis, and synaptic plasticity induced by exposure to stress [45,46,47]. These effects on EE will have a significant impact on weakening the vulnerability on adult stress.

Studies on the effects of EE in the post-weaning period have mainly focused on providing enrichment after prenatal stressful experiences, and the effects of postnatal EE on resilience to adulthood stress have not been examined. Studies of positive postnatal experiences in rats have shown that postnatal maternal care has long-term consequences for behavior and stress responses among adults [6,7,48,49]. The mechanism of this action reportedly involved DNA methylation of the hippocampal glucocorticoid receptor promoter I_7_. Adult offspring of mothers that exhibited high levels of licking and grooming showed increased hippocampal glucocorticoid receptor expression, more modest HPA responses, and lower anxiety compared with the offspring of mothers that exhibited low licking and grooming behaviors. As p11 expression is associated with depression-like states and antidepressant treatment responses, our findings show that postnatal EE’s upregulation of p11 expression through histone modification of the p11 promoter may offer a potential strategy for the prevention of adult depression. 

EE treatment in adult rodents induced epigenetic changes in DNA methylation. The offspring of parents exposed to an enriched cage environment during pregnancy exhibited significant reductions in global DNA methylation in the hippocampus and frontal cortex [50]. This finding is in line with a previous study demonstrating that EE reduced 5-hydroxymethylcytosine in the hippocampus, indicating a potential role for epigenetic markers such as DNA methylation [51]. The reduction in DNA methylation within a specific gene promoter implies an increase in the expression of that specific gene. Early-life EE may increase the expression of genes associated with the developmental programming of stress resilience by reducing DNA methylation. Moreover, exposure to EE has been shown to induce hippocampal structural changes by improving neural plasticity, as seen in increased numbers of dendritic branches and spines, enlargement of synapses, and increased neurogenesis [21]. This effect was due to increased BDNF expression via sustained histone modification of BDNF promoters (a significant increase in H3K4me3 and significant decreases in H3K9me3 and H3K27me3) in the hippocampus [52]. 

Exposure to EE has been shown to elevate serotonin concentrations and BDNF expression in the hippocampus [53,54]. One study showed that serotonin increased the p11 level only if BDNF was present, suggesting that BDNF is required for serotonin-induced increases in p11 [17]. These findings support a role for p11 in the antidepressant action of BDNF induced by EE. While the current study focused on only one gene, further studies should investigate the effects of early EE on the epigenetic regulation of stress-related genes such as BDNF. 

In the present study, both acetylation and methylation of histone H3 were altered at the p11 promoter during EE and CUS. These changes may be related to the interaction of HDAC5 with histone methyltransferases. As typical histone methyltransferases, G9a and EZH2 (specific to H3K27) and Set7/9 (specific to H3K4) should be further investigated. Moreover, it is important to show a dynamic interaction of G9a/EZH2/HDAC5 and Set7/9/HDAC5 at the promoter of p11 gene using chromatin immunoprecipitation (ChIP) assays. Further studies of current efforts are needed to provide evidence for histone acetylation and methylation-related regulation of the p11 gene.

In conclusion, the present study is the first to show prolonged exposure to early-life EE increased p11 expression through sustained chromatin regulation, which induced significant changes in histone acetylation and methylation in the hippocampus. This mechanism could partially explain interindividual difference in the stress response.

## 4. Materials and Methods

### 4.1. Animals and Groups

Pregnant C57BL/6J female mice (Daehan Biolink, Eumseong, Korea) arrived at the animal facility on the fifteenth day of pregnancy. Pregnant females were individually housed with sawdust until delivery. When parturition was observed, each dam and her offspring were carefully relocated to a standard cage. The offspring were weaned at postnatal day (PND) 21. Males are commonly used in animal experiments for depression. Data variability in females is large due to disturbing factors, such as the menstrual cycle [55]. All mice were maintained under standard laboratory conditions (21 °C, 12/12-h light/dark cycle, free access to food and water). 

All male mice were randomly assigned to four groups: Control (control group), postnatal enriched environment (EE group), adulthood chronic unpredictable stress with no prior EE (CUS group), and postnatal EE with adulthood CUS (EE + CUS group) (Figure 1). 

### 4.2. Enriched Environment (EE)

All EE animals (EE and EE + CUS group) were housed in EE cages (26 × 42 × 18 cm³) during PND 21-55, whereas other animals (control and CUS group) were housed in standard cages (20 × 26 × 13 cm³). In adulthood (8 weeks of age), all mice were housed in standard cages. The mice housed in EE cages (6–8 animals per cage) were provided with a variety of objects, such as running wheels, tunnels, small balls, and differently shaped blocks. EE cages were designed to promote neural activation, to enhance physical activity and cognition, and to provide somatosensory, visual, and motor stimulation [20]. In contrast, the control and CUS groups were housed in standard cages (6–8 animals per cage) without access to any objects. 

### 4.3. Chronic Unpredictable Stress (CUS)

The CUS and EE + CUS groups were subjected to CUS for 4 weeks starting at 8 weeks of age. All mice were housed in standard cages (20 × 26 × 13 cm³, 6–8 animals per cage). The study used a previously described procedure [56], with slight modifications. The stressors were as follows: (1) 24-h empty cage; (2) 4-h restraint stress; (3) 4-h tilt cage (45°); (4) 5-min cold swim (4 °C); (5) 1-min tail nip; (6) 24-h water and food deprivation; (7) 24-h wet cage (100 g sawdust and 200 mL water mixed together). One stressor was randomly applied daily for one week; this was repeated for 4 weeks. The control and EE groups had no contact with the stressors. 

### 4.4. Forced Swimming Test (FST)

Twenty-four hours after the last CUS protocol, the mice were exposed to the FST with minor modifications [57]. The mice were placed individually in transparent plastic cylinders (height: 25, diameter: 10 cm) that were filled to a depth of 12 cm with water (23–25 °C) for 7 min. After a 2-min habituation period, the time spent immobile was recorded during the remaining 5 min. 

### 4.5. Plasma Corticosterone Measurements

After the FST, the mice were anesthetized rapidly by Alfaxan (80 mg/kg, i.p.; Jurox, NSW, Australia) and blood was quickly collected from the abdominal aorta. Plasma corticosterone levels were measured using an enzyme-linked immunosorbent assay kit (Enzo Life Sciences, Farmingdale, NY, USA) in accordance with the manufacturer’s instructions. 

### 4.6. Measurement of mRNA Levels by qRT-PCR

The mice were anesthetized after the FST, and the whole brain was removed. The hippocampus was dissected from the brain, frozen in liquid nitrogen, and stored at –80 °C. RNA isolation, cDNA synthesis, and quantitative real-time polymerase chain reaction (qRT-PCR) were performed as described previously [38]. qRT-PCR was performed on a 7500 real-time PCR system (Applied Biosystems, Foster City, CA, USA). Glyceraldehyde-3-phosphate dehydrogenase (GAPDH) was used as a housekeeping gene. The gene-specific primers (Table 1) for p11, HDAC5, and GAPDH were used under the following conditions: 95 °C for 10 min, followed by 40 cycles of 95 °C for 15 s, 55 °C for 35 s, and 72 °C for 35 s. The cycle threshold (Ct) values were calculated automatically. Relative quantification was performed using the 2^−^^ΔΔCt^ comparative Ct method, where ΔCt = Ct_target_
_gene_ – Ct_GAPDH_. The final value was expressed relative to the control group.

### 4.7. Chromatin Immunoprecipitation (ChIP) Assays

Chromatin was isolated from the hippocampus using a standard protocol (SimpleChIP^®^ Plus Enzymatic Chromatic IP kit, Cell Signaling, Beverly, MA, USA) and immunoprecipitated with antibodies specific to histone H3 acetylated at lysine 9 and 14 (AcH3; 06-599, Millipore, Billerica, MA, USA), histone H3 trimethylated at lysine 4 (H3K4me3; ab8580, Abcam, Cambridge, MA, USA), and histone H3 trimethylated at lysine 27 (H3K27me3; ab6002, Abcam Cambridge, MA, USA) using a Simple ChIP Plus Enzymatic Chromatic IP kit in accordance with the manufacturer’s instructions. qRT-PCR was performed with primers specific to the p11 promoter region (Table 1) in the presence of SYBER Green (TOPreal^™^ qPCR 2X PreMIX, Enzynomics, Daejeon, Korea). To confirm antibody specificity, chromatin samples were immunoprecipitated with ChIP antibodies and normal rabbit IgG (#2729; Cell Signaling, Beverly, MA, USA). qRT-PCR was performed on purified DNA using a control primer set (SimpleChIP^®^ Mouse RPL30 Intron 2 Primers #7015; Cell Signaling, Beverly, MA, USA) and p11 promoter primers (Appendix A). Ct values were normalized to the input DNA. Relative quantification was performed using the 2^−^^ΔΔCt^ comparative Ct method, where ΔCt = Ct_immunoprecipitation_ − Ct_input_. The final value was expressed relative to the control group.

### 4.8. Statistical Analysis

All statistical analyses were performed using GraphPad Prism software (ver. 8.1, Graphpad Software Inc., La Jolla, CA, USA). To determine the main and interaction effects of EE and CUS, two-way ANOVA was performed. Tukey’s multiple-comparison tests were used for post hoc comparisons. *p*-values < 0.05 were considered to indicate statistical significance, and all data were presented as means ± standard error of the mean (SEM).

## Figures and Tables

**Figure 1 ijms-22-01928-f001:**
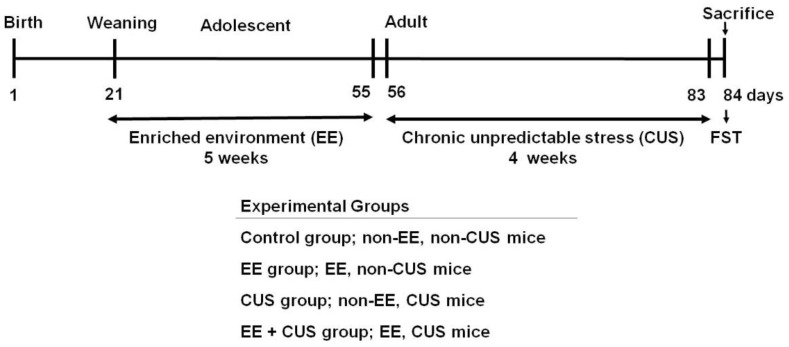
Experimental design. The mice pups were exposed to an enriched environment (EE) for 5 weeks during the post-weaning period. When the pups became adults (8 weeks of age), they were subjected to chronic unpredictable stress (CUS) for 4 weeks. Twenty-four hours after the last CUS protocol, the mice were exposed to the forced swimming test (FST). The mice were sacrificed immediately after FST.

**Figure 2 ijms-22-01928-f002:**
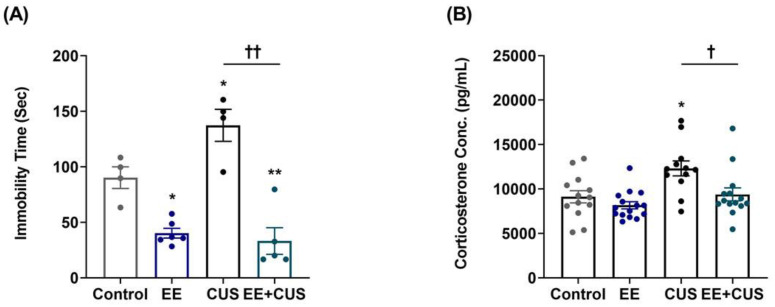
Effects of EE and CUS on depression-like behavior and plasma corticosterone levels. (**A**) Immobility time in the FST was measured at 24 h after the last CUS cession; *n* = 4–6 animals/group. (**B**) Corticosterone levels were measured using an enzyme-linked immunosorbent assay kit; *n* = 12–15 animals/group. Data are presented as mean ± SEM. * *p* < 0.05 vs. control group; ** *p* < 0.01 vs. control group; † *p* < 0.05 vs. CUS group; †† *p* < 0.01 vs. CUS group.

**Figure 3 ijms-22-01928-f003:**
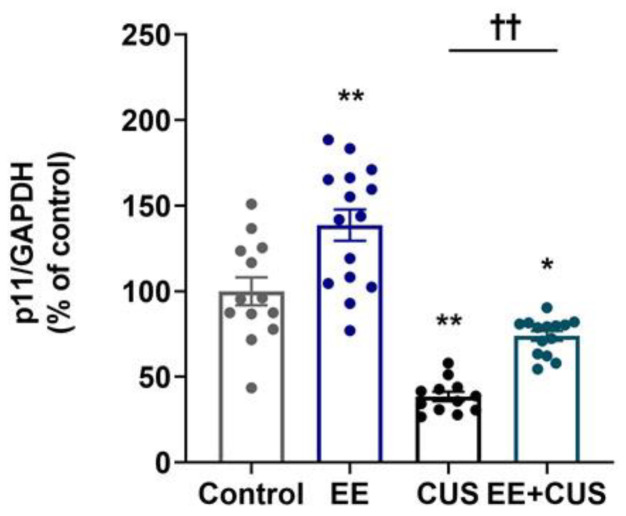
Effects of EE and CUS on hippocampal p11 mRNA levels. The p11 mRNA level in the hippocampus was measured by qRT-PCR. All quantities were normalized to glyceraldehyde-3-phosphate dehydrogenase (GAPDH). Data are expressed as values relative to the control group using the 2^−∆∆ct^ method. Data represent mean ± SEM expressed as a percentage of the control group. * *p* < 0.05 vs. control group; ** *p* < 0.01 vs. control group; ^††^
*p* < 0.01 vs. CUS group. *n* = 12–15 animals/group.

**Figure 4 ijms-22-01928-f004:**
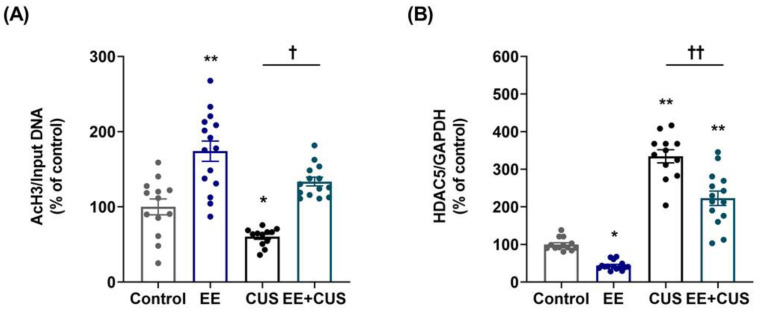
Effects of EE and CUS on the level of histone H3 acetylation at the p11 promoter and HDAC5 expression in the hippocampus. (**A**) Chromatin immunoprecipitation (ChIP) assays were performed to measure the level of acetylated histone H3 (AcH3) at the p11 promoter in the hippocampus using an antibody to AcH3. Data were normalized to input DNA. (**B**) The HDAC5 mRNA level in the hippocampus was measured by qRT-PCR. All quantities were normalized to GAPDH. Data are expressed as values relative to the control group using the 2^−∆∆ct^ method. Data represent mean ± SEM expressed as a percentage of the control group. * *p* < 0.05 vs. control group; ** *p* < 0.01 vs. control group; ^†^
*p* < 0.05 vs. CUS group; ^††^
*p* < 0.01 vs. CUS group. *n* = 12–15 animals/group.

**Figure 5 ijms-22-01928-f005:**
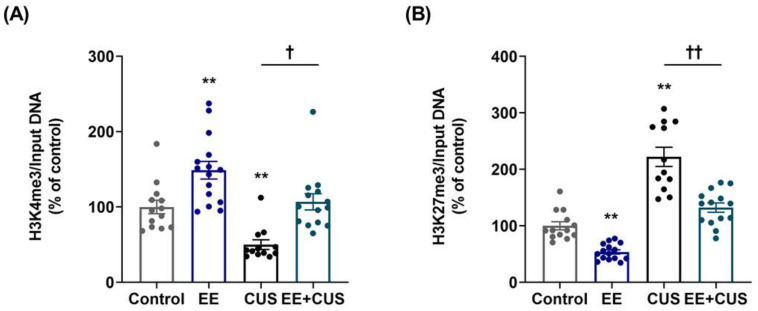
Effects of EE and CUS on the levels of histone H3 methylation at the p11 promoter in the hippocampus. The levels of histone H3 trimethylated K4 (H3K4me3, (**A**)) and K27 (H3K27me3, (**B**)) at the p11 promoter in the hippocampus were measured by ChIP assay using antibodies to H3K4me3 and H3K27me3. Data were normalized to input DNA and are expressed as values relative to the control group using the 2^−∆∆ct^ method. Data represent mean ± SEM expressed as a percentage of the control group. ** *p* < 0.01 vs. control group; ^†^
*p* < 0.05 vs. CUS group; ^††^
*p* < 0.01 vs. CUS group. *n* = 12–15 animals/group.

**Table 1 ijms-22-01928-t001:** List of primers used in this study.

	Primer Sequence (5′-3′)	
qRT-PCR for mRNA		
p11 *	Forward	TGCTCATGGAAAGGGAGTTC
	Reverse	CCCCGCCACTATTGATAGAA
HDAC5 ^#^	Forward	CCATTGGAGATGTGGAATAC
	Reverse	CAGTGGAGACAGATGTCCTT
GAPDH ^†^	Forward	AACAGCAACTCCCATTCTTC
	Reverse	TGGTCCAGGGTTTCTTACTC
qRT-PCR for histone modification		
p11 promoter ^§^	Forward	CGTTCCTCCTGCTTATCTAG
	Reverse	GCTCTTAGTATTTCAGGGCA

* Mus musculus S100 calcium binding protein A10 (calpactin) (S100a10), mRNA; NCBI reference sequence: NM_009112.2. ^#^ Mus musculus histone deacetylase 5 (Hdac5), transcript variant 4, mRNA; NCBI reference sequence: NM_001284249.1. ^†^ Mus musculus glyceraldehyde-3-phosphate dehydrogenase, pseudogene 14 (Gapdh-ps14) on chromosome 8; NCBI reference sequence: NG_007829.2. ^§^ Mus musculus strain C57BL/6J chromosome 3, GRCm38.p4 C57BL/6J; NCBI reference sequence: NC_000069.6 (GenBank Assembly ID: GCF_000001635.24).

## Data Availability

The data presented in this study are available on request from the corresponding author.

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
