# Peer review of "Early Enriched Environment Prevents Epigenetic p11 Gene Changes Induced by Adulthood Stress in Mice"

_ijms, 2021, doi:10.3390/ijms22041928_

Round 1
Reviewer 1 Report
introduction:
line 40-43: please revise the English of the sentence.
line 65-67: authors should better introduce the concept of EE, as the main treatment of the study.
Why authors chose the hippocampus? Do you consider the whole brain region? Please add this information within the text.
- 4 weeks of CUS can induce not only the despair behavior (measured by the authors with the FST) but also other deficits such as anhedonia, anxiety or cognitive deficits. Please specify why you choose this specific behavioral test.
material and methods:
why did you not consider female mice in the behavioral and molecular analysis?
How many hours after the last stressor animals were tested in the FST? when the animals were sacrificed following the FST?
in the EE and CUS section authors should better describe the housing of the animals: all the EE animals (both control and CUS) were housed in the EE cages, whereas at adulthood all mice were housed in standard cages.
Results and Discussion:
The authors indicated that EE promotes resilience in adult mice exposed to CUS via the regulation of P11. I suggest to shut down this concept throughout all the manuscript since looking at the statistical results, only in the behavioral outcome (FIG. 2A) and in the P11 mRNA levels (FIG 3) authors found a significant EE X CUS interaction, indicating that the main contribution to the results obtained derived from the EE environment per se.
- the results of the FST are obtained using 4-6 animals in each group while molecular analyses were conducted on 12-15 animals. However, it is not clear if the tested animals were included in the molecular analyses or if they represent another batch of mice. If they are included in the molecular analyses, there could be an influence of the test on corticosterone levels as well as on the p11 mRNA levels. If instead, they are a separate batch of mice, you could check if there are differences between naive and tested animals.
Line 115: authors should provide all the % of changes instead of indicated this number only for the P11 mRNA levels. Please adapt all the results section.
- the authors suggest that the underlined mechanism that prevented the depressive-like behavior in EE exposed mice and that induced the same behavior in CUS animals is based on epigenetic modifications of p11 promoter gene. To further support this assumption you could check for correlations between behavioral and molecular results.
Author Response
Summary of changes made
We appreciate the Reviewers feedback on our manuscript. All suggested corrections have been made.
Reviewer: 1
<Introduction>
1. line 40-43: please revise the English of the sentence.
→ The sentence was modified as follows (line 40-44).
<Compared to numerous studies on long-term effects of negative experiences in early life, there is limited research on the effect of positive experiences in early life. In rodents, high levels of maternal care exhibited long-term effects on the epigenomic state, brain function and behavior of offspring [6,7]>
2. line 65-67: authors should better introduce the concept of EE, as the main treatment of the study.
→ An explanation of the concept of EE as the main treatment of the study was added as follow (line 66-75).
<An environmental enrichment (EE) is the manipulation of standard laboratory conditions to reach optimization of the breeding environment by modifying the quality and intensity of environmental stimuli. EE is also known to have a significant effect on the central nervous system (CNS) at a functional, anatomical and molecular level, both in critical period in development and in adulthood [18]. And it is known that EE has an effect on neural plasticity of the brain and has been suggested the possibility of becoming a new target for the treatment of neurological and mental disorders [19]. In general, in animal experiments, EE comprises housing conditions that provide sensory, cognitive, and motor stimulation [20]. EE treatment enhanced neurogenesis and synaptic plasticity and increased BDNF expression in the hippocampus [21]>
3. Why authors chose the hippocampus? Do you consider the whole brain region? Please add this information within the text.
→ The reason for choosing the hippocampus was described as follows (line 82-85).
<According to "neurotrophic hypothesis of depression", stress-induced changes in hippocampal gene expression are known to play an important role in the pathophysiology of depression [26,27]. EE improves the neural plasticity of the hippocampus. Thus, we examined the molecular levels in the hippocampus.>
4. 4 weeks of CUS can induce not only the despair behavior (measured by the authors with the FST) but also other deficits such as anhedonia, anxiety or cognitive deficits. Please specify why you choose this specific behavioral test.
→ We think that it is not sufficient to use only FST as the outcome measure. This is a limitation of this study. This issue was described as follows (line 236-240).
<We performed the FST to assess the depression-like behavior. Although many researchers use this model for behavioral despair tests, the FST is a more appropriate tool for screening antidepressant drugs [32]. Additional behavioral tests are needed, such as the sucrose preference test to measure anhedonia and Morris water maze test to measure of spatial memory.>
<Material and methods>
1. Why did you not consider female mice in the behavioral and molecular analysis?
→ This issue was described as follows (line 344-345).
<Males are commonly used in animal experiments for depression. Data variability in females is large due to disturbing factors, such as the menstrual cycle [55].>
2. How many hours after the last stressor animals were tested in the FST? when the animals were sacrificed following the FST?
→ Twenty-four hours after the last CUS protocol, the mice were exposed to the FST. The mice were sacrificed immediately after FST. This procedure was described in detail in the legend of Figure 1.
3. In the EE and CUS section authors should better describe the housing of the animals: all the EE animals (both control and CUS) were housed in the EE cages, whereas at adulthood all mice were housed in standard cages.
→ This issue was described as follows (line 355-358).
<All EE animals (EE and EE + CUS group) were housed in the EE cages (26 × 42 × 18 cm) during PND 21-55, whereas other animals (Control and CUS group) were housed in standard cages (20 × 26 × 13 cm). In adulthood (8 weeks of age), all mice were housed in standard cages.>
<Results and Discussion>
1. The authors indicated that EE promotes resilience in adult mice exposed to CUS via the regulation of P11. I suggest to shut down this concept throughout all the manuscript since looking at the statistical results, only in the behavioral outcome (FIG. 2A) and in the P11 mRNA levels (FIG 3) authors found a significant EE X CUS interaction, indicating that the main contribution to the results obtained derived from the EE environment per se.
→ All of this concept was deleted from the manuscript (line 26-28, line 211-213, line 333-334).
The sentence on line 26-28 was replaced with: These results suggest that one of the underlying mechanisms of early-life EE may involve epigenetic modification of the hippocampal p11 gene promoter.
2. The results of the FST are obtained using 4-6 animals in each group while molecular analyses were conducted on 12-15 animals. However, it is not clear if the tested animals were included in the molecular analyses or if they represent another batch of mice. If they are included in the molecular analyses, there could be an influence of the test on corticosterone levels as well as on the p11 mRNA levels. If instead, they are a separate batch of mice, you could check if there are differences between naive and tested animals.
→ We repeated this model experiment twice. In the first experiment, the number of animals was n=4-6/group. Data of the behavior test was clearly shown in the first experiment (Figure 2A), so the behavior test was not performed in the second experiment. This is a limitation of our study. However, the second experiment also had a similar tendency to the first experiment at the molecular and corticosterone level.
3. Line 115: authors should provide all the % of changes instead of indicated this number only for the P11 mRNA levels. Please adapt all the results section.
→ Graphs of Figure 3, 4 and 5 were expressed as % of control.
4. The authors suggest that the underlined mechanism that prevented the depressive-like behavior in EE exposed mice and that induced the same behavior in CUS animals is based on epigenetic modifications of p11 promoter gene. To further support this assumption you could check for correlations between behavioral and molecular results.
→ We agree with the reviewer’s comment. To highlight this assumption, we need a correlation between behavioral and molecular results. We toned down this assumption to: One of the underlying mechanisms of early-life EE may involve epigenetic modification of the hippocampal p11 gene promoter (line 214-216).

Reviewer 2 Report
The manuscript titled 'Early Enriched Environment Prevents Epigenetic p11 Gene Changes Induced by Adulthood Stress' described the protective effect of EE in mouse model of depression and putative epigenetic mechanisms. The manuscript is well-written and the experimental design is easy to follow. Although the premise of the study is not new, the authors have found some new findings which will be important addition to the field. I have few suggestions to improve the manuscript:
- Add 'C57BL/6J' in p.1, line 21.
- Add 'in mice' at the end of title.
Author Response
Summary of changes made
We appreciate the Reviewers feedback on our manuscript. All suggested corrections have been made.
Reviewer: 2
1. Add 'C57BL/6J' in p.1, line 21.
→ ‘C57BL/6J’ is added in line 21.
2.Add 'in mice' at the end of title.
→ ‘in mice’ is added at the end of title.

Round 2
Reviewer 1 Report
The authors accomplished all my requests